# Health care professionals' experiences of screening immigrant mothers for postpartum depression–a qualitative systematic review

**Malin Skoog**[1]*, **Inger Kristensson Hallström**[1], **Andreas Vilhelmsson**[2]

1 Faculty of Medicine, Department of Health Sciences, Lund University, Sweden, 2 Faculty of Medicine, Department of Clinical Sciences, Lund University, Sweden

* malin.skoog@med.lu.se

## Abstract

### Introduction

Postpartum depression is considered a major public health problem, which immigrant mothers are at particular risk of being affected by, but it can also have long-lasting traumatic effects on the child's health and development. The Edinburgh Postnatal Depression Scale is the world's most commonly employed screening instrument for postpartum depression, used in connection with a clinical interview to screen for symptoms of postpartum depression. The aim of this study was to synthesize health care professionals (HCPs) experiences of identifying signs of postpartum depression and performing screening on immigrant mothers, since previous research suggested that this task might be challenging.

### Methods

The databases CINAHL, PubMed, PsycINFO, SocINDEX, Embase and Cochrane were searched for papers published January 2000–December 2020, reporting qualitative data on immigrants, postpartum depression and the Edinburgh Postnatal Depression Scale. Eight papers representing eight studies from four countries were included and the Critical Appraisal Skills Program was used to assess their quality. The synthesis of studies was guided by Noblit & Hare's seven-step method based on meta-ethnography.

### Findings

The synthesis resulted in two final themes: "I do my best, but I doubt that it's enough" and "I can find no way forward". The themes convey the fear and frustration that health care professionals experienced; fear of missing mothers with signs of postpartum depression, related to feeling uncomfortable in the cross-cultural setting and frustration in handling difficulties associated with communication, translated versions of the Edinburgh Postnatal Depression Scale and cultural implications of postpartum depression.

zenodo.org/record/5774074#.YbTVYC9rJQI (DOI 10.5281/zenodo.5774073).

**Funding:** The authors received no specific funding for this work.

**Competing interests:** The authors have no conflicts of interest

## Conclusions and clinical implication

By supporting HCPs' self-efficacy in handling cultural implications of postpartum depression and by developing evidence-based clinical guidelines for the use of interpreters and translated versions of the Edinburgh Postnatal Depression Scale the screening of immigrant mothers may be facilitated.

## Introduction

Postpartum depression (PPD) refers to a non-psychotic depressive episode that begins or extends into the postpartum period [1]. It is regarded as a major public health challenge for societies around the world [2,3], since PPD not only risks having long-lasting traumatic effects on the mother's health, but also on her partner's [4,5] and on the child's health and development. If left untreated, and especially if the depression lasts for a long time, it can have adverse effects on the mother-child relationship and on the child's cognitive, social and emotional development during the first years of life and cause persisting problems [6,7]. In high-income countries, PPD is the most common complication after childbearing, affecting approximately 11–14.5% of the mothers in the first three months after the delivery [8]. Mothers with a background as refugees, asylum seekers, labor or family reunification immigrants constitute a particularly vulnerable group [9]. The prevalence of PPD is estimated to be 20% for immigrant mothers as a group [10], and as much as 42% of refugees and asylum seekers [11].

From a socioeconomic perspective, the economic and humanity costs of PPD are a particular burden for society because of the detrimental effects on child health [12]. Integrating screening for PPD into primary care and providing follow-up and treatment is therefore a growing concern worldwide [13], and many countries have already implemented routine screening for PPD [14,15]. The most widely used screening tool is the Edinburgh Postnatal Depression Scale (EPDS), which is a self-report 10-item Likert-scale. It is used in connection with a clinical interview [16] and has been translated into many different languages [17]. A meta-analysis found the EPDS to have limited support, with a sensitivity of 72% and a specificity of 88%, which means that about two thirds of all depressed mothers are correctly identified [18]. Screening for PPD using the EPDS is generally found acceptable by both mothers [19] and health care professionals (HCPs) [20,21], although different aspects of administering the scale have been identified as important [22].

Studies suggest that identifying signs of PPD [23] and performing screening on immigrant mothers for the condition is challenging for HCPs [24]. Immigrant mothers are less likely than mothers belonging to the dominant population group in the country (indigenous mothers) to recall being asked about depressive symptoms and relationship problems by HCPs [23]. Moreover, they are not invited to participate in screening to the same extent as indigenous mothers [24,25] and the number of mothers identified as at risk of PPD is lower that could be expected in relation to the high prevalence in the group [25]. The quality of the screening and the validity of the EPDS when used on immigrant mothers are questioned [26]. Scales of Likert-type are known to be problematic for immigrant populations, especially for those with poor educational attainment [27]. The use of an interpreter may further affect the EPDS administration and assessment [26]. Recognition of PPD in immigrant mothers may also be problematic because of limited mental health literacy, and cultural barriers to disclosing symptoms can lead mothers to hide symptoms [28].

In light of this, it is important to provide a deeper understanding of the specific training and support needs of HCPs meeting these challenges. Previous research in the area has focused on immigrant women's perspective on handling symptoms and diagnosis of PPD and barriers to help seeking [9,28,29]. The aim of this study was therefore to synthesize health care professionals' experiences of identifying signs of PPD and screen immigrant mothers for the condition.

## Method

### Design

The meta-synthesis of this systematic review was guided by Noblit and Hare's [30] seven-phase method based on meta-ethnography and followed the eMERGe and ENTREQ reporting guidelines [31,32]. Meta-ethnography involves the systematic comparison of conceptual data from original qualitative studies to develop new overarching models and theories. The heart of meta-ethnography lies in preserving the core metaphors or concepts of the original studies, which is achieved by determining how the studies are related to each other by translating them into each other, allowing a new interpretation to emerge and thus creating a whole. Three techniques for translating the studies into each other are described according to Noblit and Hare: reciprocal, refutational and "line of argument" [30].

### Search strategy and screening

A search strategy was developed using a PEO (Populations, Exposure, Outcome) [33] structure and a systematic literature search was conducted in April 2019 in the following six databases: CINAHL, PubMed, PsycINFO, SocINDEX, Embase and Cochrane. Inclusion criteria were studies using a qualitative approach, published in English, in peer-reviewed journals between January 2000 and January 2019 and focusing on the HCPs' perspective. Articles reporting mixed-method studies were included if substantive qualitative findings were presented. Exclusion criteria were studies focusing on mothers' experiences and using quantitative data or mixed-method studies without substantive qualitative data or findings. Conference abstracts and unpublished PhD or Master's theses were not included. The literature search was updated in February 2021 to include articles published in 2019 and 2020. Key concepts were mapped to search terms, which were combined in a broad search string related to "Immigrant","Postpartum depression" and "Edinburgh Postnatal Depression Scale" (S1 and S2 Tables). Subject headings and terms, adjusted as necessary to fit each database terms, were used to both expand and focus the searches for each of the keywords using the Boolean operator 'OR'. Paired combinations of searches were made using the Boolean operator 'AND'. A hand search was carried out using the references obtained from the eligible papers.

### Data quality assessment

Articles that had met the inclusion criteria were quality-appraised using the Critical Appraisal Skills Program (CASP) tool, which consists of 10 items that allow the classification of qualitative studies within three sections referring to their methodological structure: clarity of study aims, appropriateness of methodology, ethics, clarity of design, recruitment, data collection methods and analysis, results and implications for care [34]. All studies which met the inclusion criteria and scored 6 out of 10 or above were to be included in the study.

### Data extraction and synthesis

The data extraction and synthesis were conducted according to Noblit and Hare's seven-phase method based on meta-ethnography and an overview are presented in Table 1. Since there was

**Table 1. An overview of the different phases in Noblit and Hare's seven-phase method based on meta-etnography.**

| Phase | |
|---|---|
| 1 | The research aim was articulated by all authors. |
| 2 | The material was defined by MS and AV through a search strategy. They independently screened all records by title and abstract against the inclusion criteria. Articles assessed as eligible for inclusion were furhter screened by all three authors to check for consistency. Disagreements were solved by discussion. |
| 3 | Quality appraisals was undertaken by MS and AV by comparing scores for each paper and IKH confirmed them. Differences in appraisals results in a re-read of the text until a unison decision was reached. |
| 4 | Separate careful, repeated reading of the articles by all authors. Notes of findings related to the aim (themes, metaphors, concepts with data examples and key ideas presented in results or discussion) were made in the margin and collected in a document, one for each article. Notes were compared by MS and AV and a grid of concepts for each study were jointly developed, which were confirmed by IKH. Assumptions of how studies are related to one another were made, by comparing the grid from each study to identify common and recurrent key concepts. |
| 5 | Key concepts were translated by comparing and contrasting them in relation to others in the original study and across studies. Similar initial concepts were translated into themes. |
| 6 | The translated themes were further collapsed into emerging themes named from direct quotations of the studies' participants to further communicate the findings. |
| 7 | The final themes within the synthesized findings were identified by interpreting and discussing the emerging themes to reach consensus (30). |

a significant overlap of key concepts a reciprocal translation was chosen, which means looking for similarities across studies [30].

# Results

## Search outcomes

The combined search strategy yielded in total 632 records (S1 Fig) and another record was found when hand searching references of eligible studies. After duplicates were removed (n = 287), 346 records remained. The titles and abstracts of the 346 records were screened by MS and AV, excluding another 336 records on the basis that they did not fulfil the inclusion criteria as they were either quantitative studies, systematic reviews, letter to the editors, featured immigrant mothers' experiences or did not meet the aim of the study. The hand searched record was excluded due to a publication date earlier than year 2000.The remaining nine articles, six qualitative and three mixed-methods, underwent critical appraisal. Eight of the articles received CASP scores varying from 7/10 to 10/10 and were included in the review. The most common limitation was reflection on the roles of the researchers (The results of the quality appraisal are available at https://zenodo.org/record/5774074#.YbTVYC9rJQI). One article received CASP scores below six and was excluded with reasons.

## Description of the included studies

The eight studies included represent the experiences of HCPs from four countries: UK (1) [35], Australia (3) [37,38,42], Canada (3) [36,40,41] and Sweden (1) [39]. The sample size of individual studies ranged between three and 28 participants. In six of the studies individual interviews were used and two used a combination of individual and focus group interviews. The most frequently used qualitative method of analysis was thematic analysis, followed by content analysis. Grounded theory was used in one study and an ethnographic approach in another. Key descriptive data, including authors, year, country, study aim, participants, study methodology, methods and findings from the included studies are summarized in Table 2.

**Table 2. Overview of included papers.**

| Authors, year & country | Aim | Participants | Methodology & Methods | Findings | CASP score |
|---|---|---|---|---|---|
| Griffith, 2009, [35], UK | To address the patterned and nuanced positioning of the help given to immigrant mothers in the postnatal period. | 3 participants; one clinical counsellor, one support worker and one interpreter. | Ethnographic narrative approach. Individual interviews. | Findings showed, among other things, that practitioners thought that postnatal depression has both universal and culturally specific elements. | 7/10 |
| Teng et al., 2007 et al., [36], Canada | To explore healthcare workers' experiences of providing care to recently immigrated women suffering from postpartum depression. | 16 participants working as nurses, home visitors, psychologists, psychiatrists, family doctors and social workers. | Grounded theory approach. Individual interviews using a semi-structured interview guide | Two major categories: (1) practical barriers and (2) culturally determined barriers. | 9/10 |
| Stapleton et al., 2013, [37], Australia | To identify significant emerging issues and develop recommendations for future development of care, specifically for women with a refugee background. | 18 participants: midwife, social worker, obstetrician, hospital managers, hospital interpreting coordinators, community stakeholders and research assistants. | Mixed-method approach. A thematic analysis was applied for the qualitative analysis. Four focus groups and three individual interviews. | Three key themes: (1) linguistic and cultural differences with respect to EPDS items; (2) women's vulnerabilities and protective measures concerning perinatal mental health; and (3) disclosures, concealments, and help-seeking behaviours that may impact on perinatal mental health. | 10/10 |
| Nithianandan et al., 2016, [38], Australia | To investigate barriers and enablers to implementing evidence-based, nationally recommended perinatal mental health screening, and inform sustainable implementation of a screening and referral program in women of refugee background. | 28 participants; midwives, obstetricians, maternal and child health nurses, perinatal and infant psychiatrist, perinatal mental health expert, maternity general practice liaison officer, community mental health team leader, refugee health nurse, refugee health experts, bicultural worker, interpreters. | Qualitative deductive approach. A thematic analysis applied in analysing the material from individual interviews. A semi-structured interview guide was used. | Barriers and enablers were identified and related to eight domains within the theoretical domains framework: knowledge, skills, professional roles, beliefs about capabilities and consequences, environmental context, social influences and behavioural regulation. | 9/10 |
| Skoog et al., 2017, [39], Sweden | To elucidate child health services nurses' experiences of identifying signs of PPD in non-Swedish–speaking immigrant mothers. | 13 participants: child health services nurses. | Qualitative inductive approach. Latent content analysis was applied in analysing the material from individual interviews. A semi-structured interview guide was used. | Three main categories were identified: (1) establishing a transcultural supportive relationship, (2) interpreting the mother's mood using cultural knowledge, (3) striving–sometimes in vain–when screening for PPD. | 10/10 |
| O'Mahony & Clark, 2018 [40], Canada | To increase understanding of immigrant women's reproductive mental health care services within rural settings and to inform the implementation of a cross regional research program. | 10 participants: mental health community/public health practitioners, policy makers and managers. | Mixed-method approach. A thematic analysis was applied for the qualitative analysis. An open-ended interview guide was used. | Four themes were identified: (1) community capacity building, (2) facilitators of mental health support and care, (3) barriers to mental health promotion and support, and (4) public policy and postpartum depression. | 9/10 |
| Ganann et al., 2019 [41], Canada | To explore service provider perspectives on facilitators and barriers they face in terms of providing accessible services for immigrant women with PPD. | 14 community service providers: nurses, social workers, perinatal psychiatrists, community health workers and administrators. | Content analysis was used to analyse the material from individual interviews. A semi-structured interview guide was used. | A variety of facilitators and barriers to optimal care provision were identified, in total 13 themes and 11 subthemes. A socioecological framework was used to categorize the barriers and facilitators at intra-personal, interpersonal, organizational, community, and systems levels. | 9/10 |
| Willey et al., 2020 [42], Australia | To evaluate the acceptability and feasibility of a perinatal mental health screening program for women of refugee background from the perspective of health professionals | 24 participants: midwifes, midwifery managers, bi-cultural workers, administrators, counsellors and a refugee health nurse liaison. | Mixed-method approach. A thematic analysis was applied for the qualitative analysis of two focus group and eight semi-structured interviews. | Health professionals reported improvements in identifying and referring women with mental health issues, more open and in-depth conversations with women about mental health and valued using an evidenced-based measure. Key issues included professional development, language barriers and time constraints. | 10/10 |

## Description of the themes

Following analysis, six emerging and two final themes were generated to describe HCPs' experience of identifying signs of for PPD and screening immigrant mothers. The contribution of

included studies to the different emerging themes and final themes is presented in Table 3 and examples of grid of concepts are given in the minimal dataset underlying the results is available at https://zenodo.org/record/5774074#.YbTVYC9rJQI DOI 10.5281/zenodo.5774073.

The final themes refer to the HCPs' experience of fear and frustration when identifying signs of PPD and screening immigrant mothers for it. The first theme, "I do my best, but I doubt that it's enough", reflects the HCPs' fear of missing mothers with signs of PPD and was related to feeling uncomfortable in the cross-cultural setting. This led them to doubt their ability in caring for this population at all. The second theme, "I can find no way forward", refers to the HCPs' frustration at not being able to perform the screening of immigrant mothers as well as they expected to do because of difficulties in communicating, in applying the EPDS and in the cultural implications of PPD. Despite applying proven best-practice strategies, they could not find a way forward.

**Theme one: I do my best, but I doubt that it's enough.** Theme one incorporates three emerging themes: "I was about to say become friends, but it's the wrong word"- describing the need to establish an open and trusting relationship; "My job is to put another perspective to them"–elucidating strategies in increasing health literacy, promoting mental health and identifying sings of PPD in immigrant mothers, and "You don't want to make things worse"–expressing the HCPs doubts of their own ability in identifying and supporting immigrant women with PPD.

*"I was about to say become friends, but it's the wrong word"*. Five studies [36,38,39,41,42] present findings related to HCPs' strategies to establish an open and trusting relationship with the immigrant mother, strategies which were needed in order to be able to interpret her mood all the time [38,41].

Trust was seen as the core for getting the mother to be a partner in care and to act on given advice [41]. Besides receiving the immigrant mothers with compassion, dedication and genuine desire to learn more about their culture and understand their situation, being available, receptive and responsive whenever the migrant mother sought help were emphasized as important for building trust [39].

> "that they feel that I welcome them as I welcome any fellow human being and that I don't look down on them or think that they are different, but that we . . . I was about to say become friends, but it's the wrong word . . . but that we have a good relationship. That they feel they are liked and that I listen to them. I think that's the main thing [. . .] then they trust me [. . .] that's how I think it works." [39, *p. 741*]

**Table 3. Individual study concepts as related to the emerging themes and final themes.**

| Author, year | I do my best, but I doubt that it's enough. | | | I can find no way forward. | | |
|---|---|---|---|---|---|---|
| | "I was about to say become friends" | "My job is to put another perspective to them" | "You don't want to make things worse" | "A document that non-English speaking women don't necessarily understand" | "[…] trying to communicate […] through a translator […] feels like … an exercise in futility" | "They go underground with their symptoms" |
| Griffith, (2009) [35] | | x | x | | x | x |
| Teng et al, (2007) [36] | x | x | x | x | x | x |
| Stapleton et al, (2013) [37] | | x | | x | x | x |
| Nithianandan et al, 2016 [38] | x | x | x | x | x | x |
| Skoog et al, (2017) [39] | x | x | x | x | x | x |
| O'Mahony et al, (2018) [40] | | x | x | x | x | x |
| Ganann et al, (2019) [41] | x | x | x | | x | x |
| Willey et al, (2020) [42] | x | x | | x | x | x |

Continuity of caregivers was considered as critical in order to be able to build trust, improve symptom monitoring and encourage disclosure of symptoms [36,38,42]. Longer appointments were seen as facilitating for building a relationship and thereby for the woman to feel more comfortable with discussing sensitive issues [41,42]. In two studies, it was suggested that building trust could take longer with immigrant women [39,41]. This was partly due to mothers sometimes lacking understanding of provider confidentiality and fear that it would not be maintained, especially if the HCP was a cultural peer [41]. Another reason was related to expectations of care, where immigrant women had other anticipations of care compared to indigenous women who saw it more like a matter of course [39]. Conditions for building trust were also affected by challenges with communicating through interpreters [39,41].

*"My job is to put another perspective to them"*. All studies [35–42] contributed to presenting the HCPs' strategies for identifying signs of PPD in immigrant mothers [5,34,37–39,41], increasing mental health literacy [36,42] and promoting mental health [40].

Flexibility in length of appointments was identified as a facilitator of effective assessment of migrant mothers' mood [41], which in general was considered more difficult compared to indigenous mothers because of limited mental health literacy and different views of the concept of PPD [35,39,41].

It was a common perception among the HCPs that the concept of PPD was not recognized in some cultures [35,37–39], or at least the condition was not recognized as a "medical" problem [36]. Somatic presentation of "symptoms", such as headache and back pain [35], and bodily visions of distress as static movements, looking tired and slouching, arouse suspicion among the HCPs [35,39]. Having a stiff facial expression and poor eye contact in combination with an empty sad blank look were specifically reflected on when assessing a mother's mood. Behaviors such as being a little absent and quiet, not giving information and being in a constant hurry at visits, or alternatively worried and anxious and constantly seeking help, were further interpreted as signs of PPD [39].

Acting as the "navigator" in helping immigrant mothers to become aware of the condition, of different available services, helping them to connect and also to refer them if needed was seen as an important part of the HCPs' work [41,42].

> "My job is to put another perspective to them–this is another explanation for your symptoms in terms of a psychological model of distress." [35, *p. 270*]

Any conversations about mental health were perceived as important for planting a seed in the woman's mind for future reference [42]. Interventions such as mobile applications, social media, community campaigns in local media and handing out multilingual information sheets on PPD were seen as valuable in order to increase mental health literacy in immigrant mothers and make it easier for them to disclose their feelings [36,40,41] and thereby support their help-seeking [35–39]. Mental health was further promoted by the HCPs by arranging social networking through parental program activities [39,40].

*"You don't want to make things worse"*. Six studies [35,36,38–41] contributed to describing the HCPs' feelings of lacking cultural competence and doubts in their own ability to identify and support immigrant women with PPD [36,39,41].

Working with immigrant mothers meant having to deal with complex social realties that included issues that arose from housing difficulties, spousal relationship, illness of supernatural origin and having to find a way through this to uncover a path that would help the mothers [35]. Not feeling professional in caring for this population led to fear of missing mothers with signs of PPD [36].

"it can be tough if you're not really sure that you truly understand the situation. . . you might not feel competent in a cross-cultural setting . . .you don't want to make things worse." [36, *p. 98*]

Some HCPs felt that they would benefit from training in cultural competence [38–41], while others did not think that it would help. They meant that the only solution for developing self-efficacy in their own ability to identify and support immigrant women with PPD were through clinical experience and open-mindedness [36].

**Theme two: I can find no way forward.**  Three emerging themes characterized the second theme; "A document that non-English speaking women don't necessarily understand"– describing how the HCPs adapted the challenging screening procedure to the best of their ability; "trying to communicate [. . .] through a translator [. . .] feels like . . . an exercise in futility"–elucidating lacking a common language as a major perceived difficulty, "They go underground with their symptoms"–expressing cultural understanding of PPD as a barrier and perceived as the reason for many of the challenges connected to performing effective screening.

*"A document that non-English speaking women don't necessarily understand".* Six studies [36–40,42] contributed to describing how the HCPs adapted the screening procedure to the best of their ability, but still it was perceived as challenging [39,42].

Attitudes towards performing screening with the EPDS were in general positive [38–40,42] as the EPDS allowed for more focused discussion around mental health [42] but opinions differed somewhat [36,37]. Executing the screening procedure was considered challenging and the HCPs adapted it to their best ability to make it possible for the immigrant mother to fill out the EPDS depending on her literacy level, access to a translated validated questionnaire in the mother's own language and the interpreters' skills [39]. Introducing audio versions of the EPDS to women with low literacy level were suggested for making it possible to complete the questionnaire independently [42]. It was also emphasized how the mother's educational level affected her ability to understand and fill out the EPDS [37,39,42]. The HCPs tried to normalize the screening by talking about it at earlier appointments and describing it as routine and useful [38]. Many HCPs experienced that immigrant mothers tended to interpret the statements literally [37].

"The [EPDS] is a document that non-English speaking women don't necessarily understand . . . plus when you literally translate something like: "do things get on top of you?" the literal translation is: "Is there something on top of your head?" and it's like, what's that got to do with anything? Yes, I carry water on top of my head. I carry things on my head, because that's what I do in Africa. Yes, everyday things get on top of my head." [37, *p.651*]

The HCPs were aware of that the EPDS items could be interpreted differently in different cultures and needed further explanation and clarification [38–40,42]. Concerns were raised about the cultural sensitivity and terminology of the EPDS [36,37], for example if mothers who experienced war and persecution could perceive the statements as insensitive or inappropriate [37].

Access to translated screening questionnaires was important, but the literal translations of the EPDS were in some cases considered inadequate [38,40]. Especially the Arabic and Turkish versions of the translated validated questionnaires were perceived as lacking in their translation from the original English version [39].

*"[. . .] trying to communicate [. . .] through a translator [. . .] feels like . . . an exercise in futility".* All studies described the difficulties of lacking a common language and having to rely on an interpreter as a major perceived barrier to performing effective screening [35–42].

Using an interpreter when screening for PPD was considered helpful but not ideal [37,39,41,42]. The HCPs were dependent on the interpreters' ability to translate and convey the EPDS, as well as their capacity to translate mothers' answers in a nuanced and exact way [37,39]. They found that it was hard to apply their regular conversational technique during the clinical interview [39], and having to discuss something as subtle and complicated as emotions through an interpreter aroused frustration [36].

"In my experience, trying to communicate with a woman suffering from PPD through a translator [sometimes] feels like . . . an exercise in futility." [36, *p.95*)].

The HCPs also described other circumstances related to the interpreter which could influence the screening, such as the interpreter's understanding and acceptance of the sociocultural norms within the mother's country of origin and the host country [35,37]. If the mother did not trust the interpreter's confidentiality [37–39] or the interpreter became too friendly [40] she could become apprehensive about disclosing symptoms [35–39]. Likewise, using an interpreter of the opposite sex was perceived as problematic since the mothers then might experience further stigmatization [40] and for this reason the HCPs preferred using on-site female interpreters [38,40]. Altogether the HCPs experienced an uneven quality in the interpreters' work [38,39] and standardized instructions to the interpreter regarding PPD screening were suggested [38].

*"They go underground with their symptoms".* In all studies cultural understanding of PPD was seen as a barrier and the reason for many of the challenges connected to performing effective screening [35–42].

The HCPs spoke of the stigma which existed around mental illness in many cultures, which made the mother unable or unwilling to speak about her mood and accept her own feelings [36,37,41], as did having a poor understanding of PPD [35].

"The impact is that they go underground with their symptoms. . .so then you are less likely to talk about what you're feeling." [41, *p. 194*]

The HCPs reflected on the fact that even if mothers were able to identify and accept their distress and wanted help, they felt compelled to hide it since they feared being shunned by the family and of bringing shame on the family [36–38]. As a result, the HCPs felt that mothers tended to choose "good" answers in the screening, conveying a positive message of being treated well by the husband, since "bad" answers attracted unwanted attention, implying that the husband did not fulfil his role [37]. Even if the HCPs tried to avoid it situations when the woman completed the questionnaire together with her husband occurred, if an interpreter was unavailable and she was unable to read [2].The HCPs attempted to handle the situation by normalizing mood disorders and trying to get the opportunity to perform screening without the presence of the extended family [41]. Lacking spousal support and validation of the need to seek help was highlighted as barrier to accessing care [36]. Even if the husband seemed positive to his wife wanting or needing support [37] the HCPs knew that many immigrant families kept mental ill-health in the family and did not view professional assistance as a first-line health intervention [39–40]. Another concern which the HCPs had to deal with was mothers' fear of speaking about depressive symptoms with a person in authority [36,39,40]. There was a general belief among immigrant families that the baby then would be taken away by the Child Protective Services if the parents suffered from mental illness [36].

## Discussion

This meta-synthesis of eight studies drawn from four countries represents the experiences of 126 HCPs of identifying signs of and screening migrant mothers for PPD. It paints the collective picture of identifying signs of PPD and performing the screening as complex and difficult tasks which the HCPs need support to handle.

The HCPs experienced professional frustration, which was associated with not being able to find solutions for the special challenges connected to performing screening for PPD on immigrant mothers, even though they ambitiously applied best-practice strategies; establishing a trusting relationship [36,39,41,42], practicing continuity of care, normalizing the screening [38], considering the mother's literacy level, using a translated EPDS and seeking the assistance of a skilled interpreter [39]. These strategies are in line with recommendations which emphasize that the accuracy and honesty in the screening procedure can increase when a collaborative relationship with the immigrant mother is established and the EPDS is used via an experienced interpreter. As for using the EPDS in other languages, recommendations highlight the importance of assuring oneself that the back-translation is satisfactory and that there is evidence of satisfactory face, semantic, conceptual, and technical validity [43]. The EPDS may be the most commonly used scale used to screen for symptoms of depression in the peripartum and postpartum period but there are others like for example the Postpartum Depression Screening Scale [44] and the Pregnancy Risk Questionnaire [45]. No consensus has been reached regarding which one that provides the most optimal screening in general and specifically not when applied on immigrant mothers [46].

Despite applying best-practice strategies, the screening procedure was still considered difficult [39] and difficulties related to the EPDS lacking cultural sensitivity and terminology, and the influence of different cultural values on the understanding of PPD, caused frustration among the HCPs [38–41]. Some translations also seemed inadequate, with risk of misinterpretation, and the HCPs hesitated to use them [37,39]. These difficulties were addressed in previous research and questions have been raised about the appropriateness of taking a screening instrument for PPD, developed in western societies and based on western perceptions and experiences of PPD, and applying it to culturally diverse immigrant mothers [19]. Moreover, even though the EPDS has been translated and psychometrically tested in many languages, it has not been validated for use on immigrant mothers and not via an interpreter [47]. Interpreters may struggle to find an adequate description of PPD in their own language [48]. Losing control and being reliant on the interpreters' ability not only to translate but also to convey the different items in the EPDS, which could be interpreted differently in different cultures and in need of further explanation and clarification, were therefore matters of concern for the HCPs [38–41]. Delivering the EPDS in this more guided approach is also known to affect the scores [49]. Despite consciously applying best-practice strategies and adapting as best they can the circumstances around the screening to unfamiliar cultural expressions, the use of a translated version of the EPDS, with or without an interpreter, challenged HCPs established routines and procedures, forcing them away from usual proven best practice and limiting their control. This emphasizes HCPs' need for support in handling these difficulties, showing that standardized evidence based guidelines for screening with translated EPDS and/or interpreter urgently need to be developed.

Another issue that needs to be addressed is the HCPs' fear of missing immigrant mothers with signs of PPD and not feeling professionally competent in caring for them, despite feeling a genuine commitment [36]. Professional difficulties, related to cultural competence and communication, when interacting with immigrant parents and children are known to cause frustration and time-consuming assessments of health risks among HCPs [50,51]. While some felt

they would benefit from training in cultural competence [39–41], others thought that the only solution for developing self-confidence in their own ability in performing screening of immigrant mothers was through clinical experience and open-mindedness [36]. This finding may be an expression of what has been shown in previous research, that when tested HCPs in general report a moderate level of cultural competence but are unable to fully and consistently enact this in practice [52]. In order to facilitate for HCPs to become more comfortable in cross-cultural settings, it may be valuable to offer opportunities to strengthen their self-efficacy by reflecting on how theoretical knowledge of cultural competence is translated into everyday practice. Individuals with high self-efficacy are more likely to effectively exercise control; adapting and integrating knowledge, skills and resources to manage changing situations [53]. The Knowledge, Attitude, and Self-efficacy (KAS) program is based on four components of Bandura's self-efficacy theory (i.e., performance accomplishment, vicarious experience, verbal persuasion, and physiological states) and has shown promising results when included in training programs for HCPs with the purpose of raising their self-efficacy in the identification and management of perinatal depressive symptoms in a general population [54]. In the light of this, future research may focus on developing a training program similar to the KAS program but specifically aiming at strengthen the HCPs' self-efficacy in the management of immigrant populations by targeting strategies and tools for turning cultural competence into practice, applying translated versions of the EPDS and using an interpreter in the screening.

Furthermore, the HCPs had the experience that even if mothers were able to identify and accept their distress and wanted help [36–38] they did not view professional assistance as a first-line health intervention [39,40]. This result concurs with earlier research, where immigrant women did not see health professionals and General Practicioners as an appropriate or available form of support [55]. Research conducted in low- and middle-income contexts show that psychological interventions delivered by local lay workers may be helpful in reducing material depression. The fact that such positive outcomes have been obtained in countries with relatively limited resources is promising [56] and it may also be relevant in high income countries. However, one must bear in mind that this could lead to lay workers taking on difficult cases, which they need support of HCPs in handling.

## Strengths and limitations

The methodological strengths of this paper include the use of a predetermined search strategy, quality assessment and systematic synthesis. However, despite conducting an extensive literature search in electronic databases, it is possible that not all relevant articles were located since the search was limited to articles published in English after the year 2000. In order to further strengthen the study's credibility and to make sure that all recent articles were included, the literature review was updated in June 2019, but only one more article was found to be relevant and it was included in the analysis. Because of a limited number of studies found focusing HCPs experiences of performing screening on immigrant mothers we chose to also include three studies focusing on antenatal or perinatal screening instead of solely post partum.

Since the quality of the meta-synthesis rests as much on the quality of the included original studies, quality appraisal according to CASP was undertaken, leading to the exclusion of one article. One of the included articles was written by the first author who subsequently quality appraised her own article. However the last author, who was independent from the article, quality appraised the article according to CASP reaching the same score. All three authors were involved in the identification of initial key concepts and translation of themes to reduce bias. To increase the trustworthiness, the synthesis of the data was described in detail in text and quotations were used to make the translations transparent for the reader [57].

Reflexivity is the process associated with researchers' self-awareness of how they impact and transform the research they undertake. By accounting for reflexivity, the reader is given the possibility to assess the degree to which prior views and experiences may have influenced the design, data collection and data interpretation [58]. MS and IKH had experience of working clinically in different health care contexts but only MS from the specific context. AV had no experience of working in a clinical health care context. In an attempt to avoid letting MS's pre-understanding influence the study and to strengthen the study's trustworthiness, all three authors collaborated closely during the synthesis.

## Conclusions and clinical implications

The HCPs faced many challenges, related to culture, communication and applying the EPDS when screening immigrant mothers for PPD. Despite possessing a genuine commitment and ambitiously applying proven best-practices strategies, this was not enough. Not being able to identify mothers with signs of PPD and effectively perform the screening led to feelings of fear and frustration. The challenge remains in validating and standardizing screening with a translated version of the EPDS and an interpreter to ensure its successful use with mothers of different cultural backgrounds. Evidence-based clinical guidelines for PPD screening of immigrant mothers should be developed and more studies focusing on educational interventions with the purpose of strengthening the HCPs' professional self-efficacy within this area are needed.

## Supporting information

**S1 Checklist. Skoog, PRISMA 2009 checklist.**
(DOC)

**S1 Fig. Prisma flow diagram.**
(TIF)

**S1 Table. An overview of search terms in the different databases.**
(TIF)

**S2 Table. A draft of the search strategy conducted in the electronic database PubMed.**
(TIF)

**S1 File. Skoog et al ENTREQ checklist.**
(DOCX)

## Acknowledgments

The authors acknowledge Alexandra Forsberg and Krister Aronsson (faculty librarians, Library and ICT unit, Faculty of Medicine, Lund University) for developing search strategies and conducting the search.

## Author Contributions

**Conceptualization:** Malin Skoog, Inger Kristensson Hallström, Andreas Vilhelmsson.

**Formal analysis:** Malin Skoog, Inger Kristensson Hallström, Andreas Vilhelmsson.

**Investigation:** Malin Skoog, Andreas Vilhelmsson.

**Methodology:** Malin Skoog, Inger Kristensson Hallström, Andreas Vilhelmsson.

**Project administration:** Malin Skoog.

**Resources:** Malin Skoog, Inger Kristensson Hallström.

**Supervision:** Inger Kristensson Hallström, Andreas Vilhelmsson.

**Validation:** Malin Skoog, Inger Kristensson Hallström, Andreas Vilhelmsson.

**Visualization:** Malin Skoog, Andreas Vilhelmsson.

**Writing – original draft:** Malin Skoog, Inger Kristensson Hallström, Andreas Vilhelmsson.

**Writing – review & editing:** Malin Skoog, Inger Kristensson Hallström, Andreas Vilhelmsson.

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
