## [Decision Letter · Decision Letter 0]

2 Nov 2021

PONE-D-21-10000Health care professionals’ experiences of screening immigrant mothers for postpartum depression – a meta-synthesisPLOS ONE

Dear Dr. Skoog,

Thank you for submitting your manuscript to PLOS ONE. After careful consideration, we feel that it has merit but does not fully meet PLOS ONE’s publication criteria as it currently stands. Therefore, we invite you to submit a revised version of the manuscript that addresses the points raised during the review process.

I would like to sincerely apologise for the delay you have incurred with your submission. It has been exceptionally difficult to secure reviewers to evaluate your study. We have now received two completed reviews; their comments are available below. The reviewers have raised significant scientific concerns about the study that need to be addressed in a revision.

Please revise the manuscript to address all the reviewer's comments in a point-by-point response in order to ensure it is meeting the journal's publication criteria. Please note that the revised manuscript will need to undergo further review, we thus cannot at this point anticipate the outcome of the evaluation process.

We look forward to receiving your revised manuscript.

Kind regards,

Miquel Vall-llosera Camps

Senior Editor

PLOS ONE

Journal Requirements:

2. Please ensure that you have addressed all items recommended in the PRISMA checklist including identifying the study as a systematic review in the title.

3. Thank you for stating the following financial disclosure: "No. The funders had no role in study design, data collection and analysis, decision to publish, or preparation of the manuscript. " 

5. Please note that in order to use the direct billing option the corresponding author must be affiliated with the chosen institute. Please either amend your manuscript to change the affiliation or corresponding author, or email us at plosone@plos.org with a request to remove this option.

7. Please remove all personal information, ensure that the data shared are in accordance with participant consent, and re-upload a fully anonymized data set. 

Reviewers' comments:

Reviewer's Responses to Questions

**Comments to the Author**

1. Is the manuscript technically sound, and do the data support the conclusions?

Reviewer #1: Yes

Reviewer #2: Yes

2. Has the statistical analysis been performed appropriately and rigorously? 

Reviewer #1: N/A

Reviewer #2: N/A

3. Have the authors made all data underlying the findings in their manuscript fully available?

Reviewer #1: Yes

Reviewer #2: Yes

4. Is the manuscript presented in an intelligible fashion and written in standard English?

Reviewer #1: Yes

Reviewer #2: Yes

5. Review Comments to the Author

Reviewer #1: PONE-D-21-10000

Article Type: Research Article

Full Title: Health care professionals’ experiences of screening immigrant mothers for postpartum

depression – a meta-synthesis

I read this manuscript with great interest. Postpartum depression is a major public health problem and immigrant mothers could be at particular risk. The authors present a meta-synthesis on health care professionals’ experiences of screening immigrant mothers for postpartum depression. The manuscript is well written. It highlights the fear and frustration that health care professionals experienced.

The introduction is adequate and well written.

Considering method, the section on search strategy and screening, as well as the section on data extraction and synthesis, could be shorter.

Table 1 and Table 2 could be presented as supplementary materials.

Search outcomes: One article received CASP scores below 6. What was this article about? Do the results differ from the eight other studies?

Description of the included studies: Table 3 could be shorter. For example, I think it is not necessary to present titles and all authors (first author et al) could be adequate.

Discussion: A few words on PPD management in these women could be added.

Minor remarks:

Abstract: Introduction add HCPs after health care professionals.

Discussion: First sentence I think it is eight and not seven studies.

Reviewer #2: The manuscript is well written. I have few comments:

1. Some discussions on other screening tools of depressive symptoms should come in the introduction

2. I would suggest the authors should present different phases of data extraction and synthesis in a graphical way

3. The authors should state the implication of the findings in the context of LMIC countries.

6. PLOS authors have the option to publish the peer review history of their article (what does this mean?). If published, this will include your full peer review and any attached files.

Reviewer #1: No

Reviewer #2: **Yes: **Ranadip Chowdhury

---

## [Author Response · Author response to Decision Letter 0]

13 Dec 2021

Dear Senior Editor and Reviewers,

Thank you for reviewing you manuscript entitled PONE-D-21-10000 Health care professionals’ experiences of screening immigrant mothers for postpartum depression – a systematic review and for your constructive comments for improvement. We hope you will find our attempts to develop and improve the manuscript satisfying. Please see the attached file named "Respone to Reviewers" for detailed information. 

Kind wishes,

Malin Skoog

RN PhD Student

Lund university

Sweden

---

## [Decision Letter · Decision Letter 1]

26 Apr 2022

PONE-D-21-10000R1Health care professionals’ experiences of screening immigrant mothers for postpartum depression – a systematic reviewPLOS ONE

Dear Dr. Skoog,

Thank you for submitting your manuscript to PLOS ONE. After careful consideration, we feel that it has merit but does not fully meet PLOS ONE’s publication criteria as it currently stands. Therefore, we invite you to submit a revised version of the manuscript that addresses the points raised during the review process.

 Please respond to the specific comments from Reviewers 1 and 3:Reviewer 1: the reviewer 2 suggested some discussions on other screening tools of depressive symptoms in the discussion. The authors presented this point in the introduction. I think it could be in the discussion, as reviewer 2 suggested Reviewer 3:

Suggest stating aim more clearly in the abstract. In addition, the aim needs to be clearly articulated either prior to or at the start of methods. What is written there is more an implied aim for the review.

As an overall point I wondered why the review had not included – antenatal or perinatal depression. Often screening using the EPDS etc is routine in pregnancy and many HCPs have commented on the challenges they experience. This stood out for me even more when I noted that at least three of the papers included in the review actually focus on the perceptions of HCPs who work in antenatal settings -for example the research by Nithianandan; Stapleton and Willey

I would suggest that the review should either have the broader focus or at least the intention to include studies of HCPs working antenatally with women as well as following birth. This could also be addressed in the limitations.

Introduction

In the Introduction I noted that a number of points particularly related to prevalence of PPD are referenced to quite old papers eg refs 3,4,8,9,13

Suggest all these scales need referencing here in introduction “There are specific scales developed to screen for symptoms of depression in the peripartum and postpartum period; the Edinburgh Postpartum Depression Scale (EPDS), the Postpartum Depression Screening Scale and the

Pregnancy Risk Questionnaire,”

Worth noting here “Screening for PPD using the EPDS is generally found acceptable by health care professionals (HCPs) [22, 23], although different aspects of administering the scale have been identified as important [24].” That screening also appears to be acceptable to women – Kingston is a good reference for that particularly in primary care.

I was confused by the use of term indigenous – my immediate thought was that this referred to Frist Nations populations – and I suggest that each time indigenous is used it should eb with a capital ‘I” but then I think that you may be referring to the dominant population group in a country. This needs clarifying please.

Results

I had wondered about the title of the first sub theme – referring to being a good friend… I think it is a little misleading in terms of the practice of the HCP because they correct themselves and do not emphasise becoming friends rather the quote says “…but that we have a good relationship. That they feel they are liked and that I listen to them. I…”

In the second sub theme I am reading here the importance of continuity and seeing women overt time to recognise the change in their presentation or demeanour. I think you do address this in the discussion but it is a very important point.

I wondered if more could be said about taking a different approach to screening whereby the EPDS or other tool is used as the basis of a conversation rather than just the pen and paper exercise. But this more conversational approach is probably much more relevant for the psychosocial questions that may accompany the screening using a tool.

We look forward to receiving your revised manuscript.

Kind regards,

Alessandra N. Bazzano

Academic Editor

PLOS ONE

Journal Requirements:

2. Please ensure that you have addressed all items recommended in the PRISMA checklist including identifying the study as a systematic review in the title.

Reviewers' comments:

Reviewer's Responses to Questions

**Comments to the Author**

1. If the authors have adequately addressed your comments raised in a previous round of review and you feel that this manuscript is now acceptable for publication, you may indicate that here to bypass the “Comments to the Author” section, enter your conflict of interest statement in the “Confidential to Editor” section, and submit your "Accept" recommendation.

Reviewer #1: All comments have been addressed

Reviewer #3: (No Response)

2. Is the manuscript technically sound, and do the data support the conclusions?

Reviewer #1: Yes

Reviewer #3: Yes

3. Has the statistical analysis been performed appropriately and rigorously? 

Reviewer #1: N/A

Reviewer #3: N/A

4. Have the authors made all data underlying the findings in their manuscript fully available?

Reviewer #1: Yes

Reviewer #3: Yes

5. Is the manuscript presented in an intelligible fashion and written in standard English?

Reviewer #1: Yes

Reviewer #3: Yes

6. Review Comments to the Author

Reviewer #1: The authors improved their manuscript.

The sections on search strategy and screening have been shortened.

The table has been shortened.

The text is easy to follow.

In the discussion, a section has been added on HCPs experiences of PPD management

in immigrant mothers.

All comments have been addressed. However, the reviewer 2 suggested some discussions on other screening tools of depressive symptoms in the discussion. The authors presented this point in the introduction. I think it could be in the discussion, as reviewer 2 suggested.

Reviewer #3: Thank you for the opportunity to review this paper – this is a really important issue and challenging for HCPs.

I offer the following comments to strengthen the paper.

Suggest stating aim more clearly in the abstract

In addition, the aim needs to be clearly articulated either prior to or at the start of methods. What is written there is more an implied aim for the review.

As an overall point I wondered why the review had not included – antenatal or perinatal depression. Often screening using the EPDS etc is routine in pregnancy and many HCPs have commented on the challenges they experience. This stood out for me even more when I noted that at least three of the papers included in the review actually focus on the perceptions of HCPs who work in antenatal settings -for example the research by Nithianandan; Stapleton and Willey

I would suggest that the review should either have the broader focus or at least the intention to include studies of HCPs working antenatally with women as well as following birth. This could also be addressed in the limitations.

Introduction

In the Introduction I noted that a number of points particularly related to prevalence of PPD are referenced to quite old papers eg refs 3,4,8,9,13

Suggest all these scales need referencing here in introduction “There are specific scales developed to screen for symptoms of depression in the peripartum and postpartum period; the Edinburgh Postpartum Depression Scale (EPDS), the Postpartum Depression Screening Scale and the

Pregnancy Risk Questionnaire,”

Worth noting here “Screening for PPD using the EPDS is generally found acceptable by health care professionals (HCPs) [22, 23], although different aspects of administering the scale have been identified as important [24].” That screening also appears to be acceptable to women – Kingston is a good reference for that particularly in primary care.

I was confused by the use of term indigenous – my immediate thought was that this referred to Frist Nations populations – and I suggest that each time indigenous is used it should eb with a capital ‘I” but then I think that you may be referring to the dominant population group in a country. This needs clarifying please.

Results

I think you have done an excellent job in identifying and articulating the translation of the key themes

I had wondered about the title of the first sub theme – referring to being a good friend… I think it is a little misleading in terms of the practice of the HCP because they correct themselves and do not emphasise becoming friends rather the quote says “…but that we have a good relationship. That they feel they are liked and that I listen to them. I…”

In the second sub theme I am reading here the importance of continuity and seeing women overt time to recognise the change in their presentation or demeanour. I think you do address this in the discussion but it is a very important point.

The discussion covers important issues and addresses the findings

I wondered if more could be said about taking a different approach to screening whereby the EPDS or other tool is used as the basis of a conversation rather than just the pen and paper exercise. But this more conversational approach is probably much more relevant for the psychosocial questions that may accompany the screening using a tool.

7. PLOS authors have the option to publish the peer review history of their article (what does this mean?). If published, this will include your full peer review and any attached files.

Reviewer #1: No

Reviewer #3: No

---

## [Author Response · Author response to Decision Letter 1]

9 Jun 2022

Reviewer 1 - comments 

1. Reviewer 2 suggested some discussions on other screening tools of depressive symptoms in the discussion. The authors presented this point in the introduction. I think it could be in the discussion, as reviewer 2 suggested. 

Reply: Thank you noticing this. The sections on other screening tools of depressive symptoms have been moved to the discussion.

Reviewer 3 - comments 

1. Suggest stating aim more clearly in the abstract. In addition, the aim needs to be clearly articulated either prior to or at the start of methods. What is written there is more an implied aim for the review. 

Reply: Thank you for your constructive comment. The aim has been clarified in both abstract and introduction. The abstract now reads “The aim of this study was to synthesize health care professionals (HCPs) experiences of identifying signs of postpartum depression and performing screening on immigrant mothers, since previous research suggested that this might be challenging” and the introduction “The aim of this study was therefore to synthesize health care professionals’ experiences of identifying signs of PPD and screen immigrant mothers for the condition”.

2. As an overall point I wondered why the review had not included – antenatal or perinatal depression. Often screening using the EPDS etc is routine in pregnancy and many HCPs have commented on the challenges they experience. This stood out for me even more when I noted that at least three of the papers included in the review actually focus on the perceptions of HCPs who work in antenatal settings -for example the research by Nithianandan; Stapleton and Willey. I would suggest that the review should either have the broader focus or at least the intention to include studies of HCPs working antenatally with women as well as following birth. This could also be addressed in the limitations.

Reply: Thank you for noticing this. A section was added under limitation explaining that ”Because of a limited number of studies found focusing HCPs experiences of performing screening on immigrant mothers we chose to include three studies focusing on antenatal or perinatal screening instead of solely postpartum.

3. In the Introduction I noted that a number of points particularly related to prevalence of PPD are referenced to quite old papers eg refs 3,4,8,9,13 

Reply: Thank you for pointing this out. Reference 3 and 4 has been replaced and reference 8 and 9 has been removed. Reference 13 was kept since it is an important reference for arguing the relevance of the paper. 

4. Suggest all these scales need referencing here in introduction “There are specific scales developed to screen for symptoms of depression in the peripartum and postpartum period; the Edinburgh Postpartum Depression Scale (EPDS), the Postpartum Depression Screening Scale and the

Pregnancy Risk Questionnaire. 

Reply: Thank you noticing this. References had been added for the different scales. 

5 Screening also appears to be acceptable to women – Kingston is a good reference for that particularly in primary care. 

Reply: Thank you for pointing this out. The suggested reference Kingston was not retrieved, and another reference was therefore used. 

6. I suggest that each time indigenous is used it should eb with a capital ‘I” but then I think that you may be referring to the dominant population group in a country. This needs clarifying please. 

Reply:Thank you for pointing this out. The term indigenous has been clarified. 

7. Results. I think you have done an excellent job in identifying and articulating the translation of the key themes

I had wondered about the title of the first sub theme – referring to being a good friend… I think it is a little misleading in terms of the practice of the HCP because they correct themselves and do not emphasise becoming friends rather the quote says “…but that we have a good relationship. That they feel they are liked and that I listen to them. I…”

In the second sub theme I am reading here the importance of continuity and seeing women overt time to recognise the change in their presentation or demeanour. I think you do address this in the discussion but it is a very important point. 

Reply: Thank you so much for your positive comment. The title of the first subcategory has been changed to; “I was about to say become friends, but it’s the wrong word” to avoid misleading the reader. 

8. The discussion covers important issues and addresses the findings. I wondered if more could be said about taking a different approach to screening whereby the EPDS or other tool is used as the basis of a conversation rather than just the pen and paper exercise. But this more conversational approach is probably much more relevant for the psychosocial questions that may accompany the screening using a tool. 

Reply: Thank you for raising this interesting point. This is an important question, but we feel that it belongs in another context were it can be given the space it deserves.

---

## [Editor Report · Decision Letter 2]

23 Jun 2022

PONE-D-21-10000R2Health care professionals’ experiences of screening immigrant mothers for postpartum depression – a systematic reviewPLOS ONE

Dear Dr. Skoog,

Thank you for submitting your manuscript to PLOS ONE. After careful consideration, we feel that it has merit but does not fully meet PLOS ONE’s publication criteria as it currently stands. Therefore, we invite you to submit a revised version of the manuscript that addresses the points raised during the review process.

The authors have done an excellent job of addressing reviewer's comments. There is an important final revision needed after which the paper will be accepted, the authors should revise the title of their manuscript to describe the study as a "qualitative systematic review" and cite the ENTREQ reporting guidelines in the Methods section, along with completing the ENTREQ checklist. See: https://www.equator-network.org/reporting-guidelines/entreq/ There is also a minor change needed, there appears to be a grammatical error in Table 3 where the word "women" should replace "woman", or else indicate this was the verbatim quote using [sic].Please ensure that your decision is justified on PLOS ONE’s publication criteria and not, for example, on novelty or perceived impact.

We look forward to receiving your revised manuscript.

Kind regards,

Alessandra N. Bazzano

Academic Editor

PLOS ONE
---

## [Author Response · Author response to Decision Letter 2]

27 Jun 2022

Dear Reviewer

We hop you find the changes to improve the manuscript PONE-D-21-10000R2 Health care professionals’ experiences of screening immigrant mothers for postpartum depression – a qualitative systematic review satisfying.

Kind regards

Malin Skoog,

RN, PhD

Lund University

---

## [Editor Report · Decision Letter 3]

29 Jun 2022

Health care professionals’ experiences of screening immigrant mothers for postpartum depression – a qualitative systematic review

PONE-D-21-10000R3

Dear Dr. Skoog,

We’re pleased to inform you that your manuscript has been judged scientifically suitable for publication and will be formally accepted for publication once it meets all outstanding technical requirements.

Kind regards,

Alessandra N. Bazzano

Academic Editor

PLOS ONE
---

## [Editor Report · Acceptance letter]

5 Jul 2022

PONE-D-21-10000R3 

Health care professionals’ experiences of screening immigrant mothers for postpartum depression – a qualitative systematic review 

Dear Dr. Skoog:

I'm pleased to inform you that your manuscript has been deemed suitable for publication in PLOS ONE. Congratulations! Your manuscript is now with our production department. 

Kind regards, 

on behalf of

Dr. Alessandra N. Bazzano 

Academic Editor

PLOS ONE